# Compressed Air-Driven Continuous-Flow Thermocycled Digital PCR for HBV Diagnosis in Clinical-Level Serum Sample Based on Single Hot Plate

**DOI:** 10.3390/molecules25235646

**Published:** 2020-11-30

**Authors:** Kangning Wang, Bin Li, Wenming Wu

**Affiliations:** 1State Key Laboratory of Applied Optics, Changchun Institute of Optics, Fine Mechanics and Physics, Chinese Academy of Sciences, Changchun 130033, China; WKN19950104@163.com (K.W.); libinsemail@163.com (B.L.); 2University of Chinese Academy of Sciences, Beijing 100049, China

**Keywords:** continuous-flow PCR, compressed air-driven micropump, diagnosis in clinical-level serum, digital PCR

## Abstract

We report a novel compressed air-driven continuous-flow digital PCR (dPCR) system based on a 3D microfluidic chip and self-developed software system to realize real-time monitoring. The system can ensure the steady transmission of droplets in long tubing without an external power source and generate stable droplets of suitable size for dPCR by two needles and a narrowed Teflon tube. The stable thermal cycle required by dPCR can be achieved by using only one constant temperature heater. In addition, our system has realized the real-time detection of droplet fluorescence in each thermal cycle, which makes up for the drawbacks of the end-point detection method used in traditional continuous-flow dPCR. This continuous-flow digital PCR by the compressed air-driven method can meet the requirements of droplet thermal cycle and diagnosis in a clinical-level serum sample. Comparing the detection results of clinical samples (hepatitis B virus serum) with commercial instruments (CFX Connect; Bio Rad, Hercules, CA, USA), the linear correlation reached 0.9995. Because the system greatly simplified the traditional dPCR process, this system is stable and user-friendly.

## 1. Introduction

Polymerase chain reaction (PCR) amplification technology has been proposed for 37 years. During this period, PCR has developed into a key technology and conventional technology in the field of molecular biology, which has greatly promoted the development of various fields of life sciences [1,2,3,4,5,6,7]. As an emerging technology, microfluidic technology has shown great advantages in molecular research. With the rapid development over the past decade, digital PCR (dPCR) has evolved from the combination of PCR amplification technology and microfluidic technology, which is used to detect disease. This has brought biomedical research into a new stage [8,9,10,11].

At present, PCR technology is mainly divided into three stages. The first generation of PCR technology was the original PCR. The target gene was amplified by an ordinary PCR amplification instrument. Then the products were analyzed by agarose gel electrophoresis, and only qualitative analysis was performed. The second generation of PCR is real-time PCR, also known as qPCR, which monitors the accumulation of amplification products by adding fluorescent probes to the reaction system to indicate the process. The results can be judged by fluorescence curve, and can be quantified by CT value and standard curve. The third generation of PCR technology is called digital PCR (Digital PCR), which is a new PCR detection method at present. It can detect and quantify the nucleic acid. By directly counting the target molecules without relying on any calibrator or appearance, the absolute number of target molecules to be detected can be determined as low as a single copy. Digital PCR is classified according to the method of sample dispersion, mainly divided into: droplet-based and microwell-based and chamber-based methods. Microwell-based PCR generates microdroplets by microfluidic technology, in which microdroplets are uniformly introduced into the micro-reaction cell of the plate-shaped chip including proposed methods using liquid marbles, core shell beads, etc. [12,13,14]. The DNA fragments are amplified by setting different temperatures and durations [15,16,17,18,19]. Droplet-based dPCR uses microfluidic technology to wrap the water phase in the oil phase to form droplets. At the same time, each droplet is a separate reaction chamber, and then the droplets pass through different temperature bands to amplify the DNA fragments [20,21,22,23,24]. The latter does not require repeated heating or cooling of the PCR device, and the heating and cooling rates are generally not limited by the heat capacity of the PCR device system, and the reaction speed is faster [25,26,27].

In a continuous-flow PCR device, the driving force plays an important role in the generation and flow of droplets. However, the driving force of most devices is provided by external pumping devices, some of which are mechanical syringe pumps, and some rely on computer-programmed pressure controllers to provide injection pressure [25,26,27]. However, these devices are mostly bulky and expensive. The pumping device that relies on the controller, while accurate in pumping control, requires complex experimental devices and professional operators [28,29,30,31,32,33]. Therefore, the compressed air-driven method has been proposed and applied to a continuous-flow PCR of multiphase fluid [34]. However, these systems are limited to a single fluid (both conventional PCR and real-time PCR) and do not separate the reagent into small droplets, which does not meet the requirements of digital PCR [35].

This paper presents a compressed air-driven, user-friendly, and stable continuous-flow dPCR reaction system. This system upgrades the real-time quantitative PCR to digital PCR, which upgrades the relative quantification based on the amplification curve to absolute quantification. A combination of a manual syringe and a microfluidic tube was used to create a uniform droplet. Such a device integrates droplet formation, collection, and reaction with virtually no complicated experimental setup, minimizing system volume. During the experiment, the droplet formation was very stable without the use of a surfactant, and the flow rate of the droplets was also very stable. The droplets are stable in long-distance transport and do not break apart even in high temperature environments. Image data acquisition is performed on the entire cycle using CMOS, and the product of each cycle is clearly understood. We independently developed a software system for automatic analysis of micro-droplets, which realized the automatic analysis of data. Experiments showed that this compressed air-driven continuous-flow dPCR reaction system had comparable amplification efficiency for commercial digital PCR machines. Therefore, the compressed air-driven continuous-flow dPCR reaction system is convenient to use, especially for untrained users.

## 2. Methods

### 2.1. Composition of Micro Devices

The assembly of micro-devices is the basis for the generation of micro-droplets. As shown in Figure 1, the device consists of two syringes, two 34 g fine needles, a Teflon microtube with a diameter of 0.16 mm × 0.32 mm (MAFLONR, ASONE, Shanghai, China), a quartz tube with a diameter of 50 μm and a waste liquid collection bottle. The function of the continuous-flow dPCR experimental device was realized by the following: Firstly, two 34 g fine needles were combined to form a microchannel with parallel tubes and uniform holes as a droplet generation device. Secondly, the parallel end of the combined fine needle head was connected with the Teflon micro pipeline to form a droplet transmission pipeline. Finally, several disposable syringes were selected as the storage device of oil and water phases, and a compressed air-driven micropump composed of syringes with water and oil phases was designed. A compressed air-driven continuous-flow dPCR experimental device was assembled by droplet generation, droplet transport and compressed air-driven micropump. A compressed air-driven micropump can provide power for the flow of reagents. The air in the syringe was compressed to provide power for the whole system, which is measurable and stable. The Teflon microchannel for droplet transport was wound on a special shape mandrel made of PDMS and placed on the top of the heating platform to ensure the temperature required for amplification. The temperature of the heater was set as the reaction temperature in the high temperature zone, and the temperature of the top layer of the mandrel was reduced to the reaction temperature in the low temperature zone due to the thermal conductivity of PDMS and the natural convection of air. For the excitation light source of dPCR liquid drop fluorescence, the LED array (xpe60w, Cree, NC, USA) with the luminous frequency of 470 nm was used, and the narrow-band filter (470–30 nm, Xintian bori, Beijing, China) was used for filtering.

### 2.2. Droplet Generation

We present a new method to generate droplets based on our previous studies, using a disposable syringe to provide the pressure, the bent thin wire was used to fix the piston of the syringe so that its position does not change [36]. A 10 mL disposable syringe and a 5 mL disposable syringe (Xuancheng Jiangnan Medical Devices Co., Ltd., Xuancheng, China) were connected to the quartz tube through the microchannel, and the reagent flow rate can be controlled by changing the length of the quartz tube. The needle attached to the syringe is the key to generating microdroplets, and the needles are parallel to each other (Figure 1). During actuation, the air inside the fluidic conduit is compressed by two syringe pistons, producing a conduit pressure that is higher than the atmospheric pressure. The water phase and the oil phase were loaded separately into the two disposable syringes, respectively. Since the end of the microchannel is outside atmospheric pressure, the reagent was pumped to the droplet generating device, which produces droplets that will be delivered to different viewing areas.

### 2.3. Reagents

The 50 μL dPCR reaction mixture contained 18 μL HBV-PCR reagent, which was supplied by NEPG Liaoning Biopharmaceutical Co., Ltd. The 8 μL reaction mixture was injected into an aqueous phase syringe, and then the DNA was subjected to thermal cycle amplification and detection using a designed dPCR system. The entire cycle was carried out 42 times. In order to ensure that the reaction proceeded smoothly, we used an infrared imager to detect the temperature of the thermal cycle module. The final test results are as follows: the amplification ambient temperature was 94.2 °C (±0.5 °C) for 10 s and 59.3 °C (±0.5 °C) for 30 s.

### 2.4. Image Acquisition and Processing

The fluorescence detection system used a 3W high-power LED as the fluorescence excitation light source, and 480 nm filter was installed in front of the excitation light source to ensure that it can excite fluorescence, but it will not generate excessive noise. The fluorescence receiving device uses CMOS with 20 million pixels. The front end of CMOS was equipped with an aoptical lens and a 520 nm filter to receive the fluorescence signal and display the real-time image on the PC connected with it. Because CMOS can adjust the focal length according to the need, this structure can monitor the fluorescence of each cycle of the amplification reaction in real-time. The fluorescent droplet counting software installed on the PC calculated and counted the brightness of each droplet by playing a video on the software interface, and then drew the brightness map of the droplet through the derived Excel table to get the linear relationship.

## 3. Results and Discussion

### Fabrication of Microdevice

Previous continuous-flow PCR systems relied on electric pumps to generate droplets that provide propulsion for the entire system. The compressed air-driven micropump can still ensure that the PCR proceeds normally without providing electrical energy and can produce droplets of stable size. In this continuous-flow digital PCR system, a compressed air-driven micropump is an important device. In order to test whether the compressed air-driving micropump meets the requirements for use, we monitor its own pressure changes and droplet generation and flow conditions.

The difference in air pressure between the cavity in the compressed air-driven micropump and the outside atmospheric pressure is the only source of propulsion for the entire system, so it is very important to monitor the pressure. For this purpose, three groups of experiments were carried out with different initial pressures, and the pressure changes were recorded (Figure 2a). Here, we call the pressure from low to high as low pressure, medium pressure and high pressure. In each group of experiments, the reagent amount is the same. The tail end of Teflon microchannel is connected with a quartz tube with an inner diameter of 50 μm and a length of 120 mm. The cavities of oil phase and water phase are independent and connected with the pressure gauge separately, so that the pressure value can be recorded at any time. Obviously, under the same amount of reagent, the greater the pressure, the shorter the flow time of the solution, and the pressure difference between the oil phase and water phase is very small in the same group of experiments. Among the three initial pressures, the pressure decays faster under high pressure.

In order for the reagent to react smoothly in the chip, a suitable flow time is necessary. As shown in Figure 2b, we compared the droplet flow times for different initial pressures and different quartz tube lengths. During the entire experiment, the time from the flow of the reagent to the first cycle was started, and the time was stopped until the 40th cycle. Experiments show that when the initial length of the quartz tube is 12 cm, the initial pressure is 60 kPa, the reagent flow time is the longest, about 2700 s. When the initial length of the quartz tube is 6 cm, the initial pressure is 80 kPa, the difference in flow time is about 2000 s. The flow time under the difference is about 2000 s. The droplet flow time was continuously studied using the control variable method. In the experiment, the initial pressure and the length of the quartz tube were controlled as the only variables, and the other conditions were unchanged, and the flow time of the reagent under different conditions was obtained. According to the amplification conditions of the selected reagent, a quartz tube with a length of 12 cm was finally selected as a tail tube and connected to the end of the Teflon microtube. The initial pressure is 60 kPa.

The microdroplet generation experiment was carried out using the above conditions. As shown in Figure 2c, the flow velocity of the droplets in the microchannel was stable, the slowest flow rate was 1.2 mm/s, and the fastest flow rate was 1.4 mm/s. The flow rate difference is only 0.2 mm/s. The above data prove that the droplets in our system can flow smoothly. The smooth flow rate is an important prerequisite for the success of the experiment. The droplets generated by the droplet generation system are generally close to each other. If the flow rate is unstable, the droplets collide and squeeze each other, which will greatly improve the droplets. It is likely that once the droplets are dissolved, the system loses the function of digital PCR. However, only a stable flow rate is not enough, and the size of the droplet itself containing the gene to be measured and the droplet spacing also plays an important role. The Teflon micro-pipe used in the experiment was stretched to four times its original length, and its inner diameter was 0.15 mm. For the convenience of observation, the ink was used to test the droplets instead of the reagent. As shown in Figure 2d, the droplets flowing in the pipe are very uniform in size and have a length of 0.5 mm. After calculation, the droplet volume is about 8.83 nL. In addition, the droplet pitch generated in the figure is stable, about 3 mm. The above results prove that the compressed air-driven device of the system is sufficient for the reagent propulsion work of the whole system, and the operation of the droplet completely satisfies the amplification conditions.

Three different concentrations of the same reagent were selected as samples for the sample concentrations of 10^3^, 10^4^ and 10^5^ International Unit/mL (IU/mL), respectively. Amplification curves were obtained by placing the samples to be tested in a commercial qPCR instrument (Figure 3a). The same samples as from the above experiments were configured and passed to our system for the reaction. During the reaction, the droplet status was observed and the video was recorded in CMOS. After the completion of the reflection, the video analysis team developed the fluorescence analysis software to perform fluorescence signal statistics, and output a dot pattern of droplets’ brightness (Figure 3b). Finally, the initial number of DNA of the sample was calculated using the fluorescence signal data. The initial number of DNA in the 10^5^ IU/mL sample was calculated to be 18.16 copy, the initial number of DNA in the 10^4^ IU/mL sample was 127.92 copy and the initial number of DNA in the 10^5^ IU/mL sample was 857.00 Copy. Then, the logarithm of the initial number of DNA was taken as the ordinate and the CT value in the amplification curve, generated by the conventional PCR instrument, was draft as the abscissa. The standard curve (Figure 3c) is presented and the respective amplification is performed. The correlation coefficient R^2^ of the standard curve is calculated to be 0.9981. An image of real-time fluorescence of droplets taken by a CMOS camera is shown in Figure 3d.

## 4. Conclusions

In short, we propose a continuous-flow dPCR system, which greatly simplifies the operation process. It is also the first reported system in the world that uses the spontaneous transmission of fluid and a single constant temperature heater. The system has successfully realized the absolute quantitative detection of HBV serum samples. Firstly, the use of single gene wrapping technology to generate microdroplets is the first step in the overall reaction and the basis for the success of the reaction. Experiments have shown that the droplets generated by our system fully meet the requirements of digital PCR. This is followed by a drive that powers the generation and flow of droplets. A stable and suitable driving force produces perfect droplets, which is one of the highlights of this paper. The above experimental data show that the propulsion force provided by the compressed air-driven device can produce micro-droplets with a uniform flow rate. More importantly, such a compressed air-driven device completely utilizes the air pressure difference to provide energy without consuming electric energy, and the volume is much smaller than the syringe pump. In terms of thermal cycling, in order to make the whole system as small as possible, we used a single heat source heating method to perform gene amplification using a two-step method. The experimental results show that the PCR using the two-step method is successful, and the system can completely rely on a single heat source to complete the temperature cycle. The above conclusions all indicate that the compressed air-driven continuous-flow digital PCR system is stable and user-friendly.

## Figures and Tables

**Figure 1 molecules-25-05646-f001:**
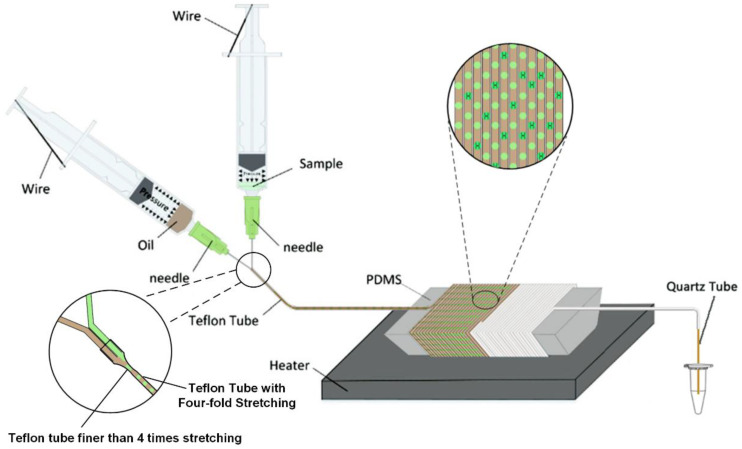
Schematic of the micro device.

**Figure 2 molecules-25-05646-f002:**
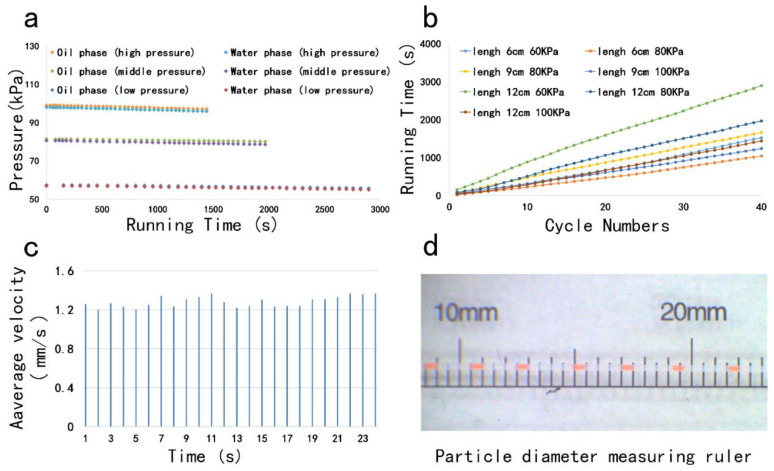
(**a**) Pressure changes in air chambers of water and oil phases under different initial pressures; (**b**) the influence of the length and initial pressure of the tailpipe on the flow time of droplets; (**c**) flow rate of droplets; (**d**) measurement of droplet size and spacing.

**Figure 3 molecules-25-05646-f003:**
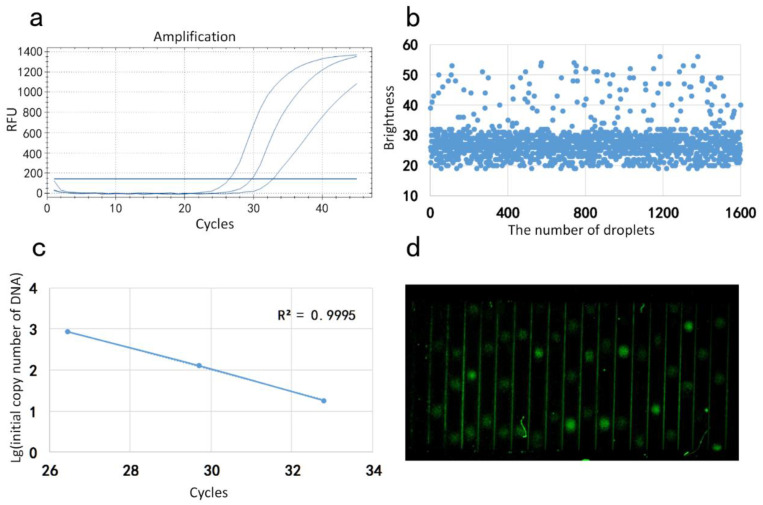
(**a**) Standard curve of reagents measured by our device; (**b**) statistical result of droplets’ brightness; (**c**) amplification curve of the reagents measured by commercial qPCR (BIO-RAD CFX Connect); (**d**) the image of real-time fluorescence.

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
