# Peer review of "Compressed Air-Driven Continuous-Flow Thermocycled Digital PCR for HBV Diagnosis in Clinical-Level Serum Sample Based on Single Hot Plate"

_molecules, 2020, doi:10.3390/molecules25235646_

Round 1

Reviewer 1 Report

The manuscript describes a self-driven, continuous flow, digital microPCR using one heater and real-time monitoring of the DNA amplification. This reviewer recommends transfer of this manuscript to “mdpi Micromachines”, as it is more appropriate for this work than “mdpi Molecules”.

The group has published several articles on similar self-driven microPCRs, therefore they do not go into details which are necessary for the reader to understand the work independently.

  • For example, the authors state that the mPCR performs a 2T protocol, at 92C for 10s and at 59 C for 30 s. How was this residence time determined at each temperature?
  • 42 cycles were also performed, based on the number of turns the Teflon tube is coiled around the PDMS slab.
  • Why the conditions described in lines 185-187 were finally chosen for the amplification? Amplification was not possible at shorter times?
  • Why does the fluorescence signal (Fig. 3b) vary between 20 and 50 units? What does the high value correspond to?        

Last, many syntax errors exist in the manuscript, please revise the manuscript. For example, revise the sentence (“Use a combination…) in lines 70-71. Line 80, “has comparable amplification efficiency with ..”. Line 88, revise the sentence “In order to …”. Line 170, do you mean “the shorter the flow time”? Lines 178-180, what is the length of the quartz tube 12 or 6 cm (both are written in the same sentence)? Line 181: please write “the difference in flow time is about 2000 s”. Revise the sentence in lines 215-216. Revise the sentence in line 222 “105 IU/ml …”. Line 41, what is CQ? Line 51 “the water phase in the aqueous phase”?  

Reviewer 2 Report

This article describes a continuous-flow droplet PCR in a capillary wrapped around a PDMS block or mandrel such that the droplets experience different temperatures on opposite sides of the mandrel. They monitor fluorescence in the droplets in 'real time' by imaging the flowing droplets on the top side of the mandrel.

Ideas like this have been around for a long time; see for example Dorfman, Anal Chem 2005, https://pubs.acs.org/doi/abs/10.1021/ac050031i (although that's without fluorescence detection) or Kiss, Anal Chem 2008, https://www.ncbi.nlm.nih.gov/pmc/articles/PMC2771884/ (although in microfluidic chip). The rationale here for wanting to do "real-time" imaging on droplets isn't clear. Whether chip-based or droplet-based, the principle behind digital PCR is simply to look at reaction endpoints to "count" number of templates at limiting dilution such that droplets have zero or one copies. The kinetics of how the droplet got there is not interesting. So in the author's approach, why not just have a single-point detector measuring the droplet fluorescence in single-file?

Much of the focus here is on the "self driving micropump", and the control of flow rate via a capillary tube "tailpipe" as a flow restrictor on the back end. Strictly speaking a self-driving micropump is an impossibility. The driving force for flow is having an air gap in the oil and sample syringes, compressing that with the plunger, and then locking the plunger in place. That isn't really "self driving". A concern I have is that if they're not careful about keeping relatively uniform pressure between the sample and oil syringes, they could end up with fluid from one syringe back-flowing into the other syringe. A second concern is that this initial setup of jamming two bent fine needles into a teflon capillary tube looks very difficult if not dangerous, and such a connection is not immune to leakage if the pressure is too high. Why not use something like a commercial T fitting for microtubing?

Overall the English of this manuscript is very poor, and this may contribute to my not fully understanding what they did or why they did it. It would need complete re-writing for comprehensibility. Figure 3 also needs better explanation. It seems like there was a qPCR experiment, and a droplet experiment, but it is unclear how these connect to each other. There's a rather wordy description of how they constructed a qPCR standard curve (lines 219-226) but I didn't see an attempt to relate that quantitation to whatever quantitation they calculated from their droplets. Figure 3 should also show results for a no-template control, as this would help in their thresholding for what is positive, what is negative. And it still remains unclear, are they tracking droplet brightness at a particular location or are they somehow looking at droplet intensity across the field of view (i.e. in multiple lengths of the tubing as seen in Fig 3d)? Does it matter if they can track with droplet is which on each turn around the mandrel?

Overall I would recommend resubmitting this after a complete re-write for clarity, and better explanation of why they take this approach, how it compares to conventional ddPCR as well as the prior work in the field of continuously flowing droplet PCR, and strengths and weaknesses. Also please don't use the term "self-driving micropump". If taken literally this violates the laws of thermodynamics as it would be a form of perpetual motion machine.

Reviewer 3 Report

The authors presented thermal cycled digital pcr for hbv diagnosis on single plate. They demonstrated good results in the paper. The manuscript is organised well. The clarity of presentation is average. The authors could improve the paper significantly y adding some minor details. Some of the oservations are listed below.

  1. Please add more details in the introduction section particularly about recent developments in digital PCR tchniques( including proposed methods using liquid marbles, core shell beads etc). Some references are given elow.
    1. https://pubs.rsc.org/en/content/articlelanding/2018/lc/c8lc00990b#!divAbstract
    2. https://pubs.rsc.org/en/content/articlelanding/2019/lc/c9lc00676a#!divAbstract
    3. https://www.mdpi.com/2072-666X/11/3/242
  2. Please provide some details about the fluorescence counting softare used
  3. The authors have not mentioned about the droplet volume control. How to estimate or detect the volume inconsistencies of droplets.
  4. The compressibility of fluid inside the syringe, friction between syringe cavity and piston etc can affect the droplet volume. Is there any feed back mechanism to ensure the droplet volume inconsistencies. 
  5. Droplet volume inconsistencies can greatly affect the number of template dna in one droplet. This will greatly affect the accuracy of dpcr in copy number detection and all. Please describe your method to avoid this problem. 
  6. Line number 243 . its not droplet size its droplet volume. 
  7. The authors have not discussed much about the heater. How accurate the temperature are. 
  8. Did you perform any test to ensure the temperature offered by the heater.
  9. what is the ramping rate of the heater used as it affect the efficiency of dpcr
  10. I am not seeing any details of clinical samples used ( Please explain whether used any clinical sampes. If yes provide etchical clearance. Title mentions "clinical-level serum" . If not consider revising the title. ) 

Reviewer 4 Report

This manuscript describes compressed air-driven continuous-flow digital PCR system using a 3D microfluidic chip.  Authors demonstrated that the droplets generated by the system meet the requirements of digital PCR.

This method is interesting and useful in this field.  This kind of micropump system could be user-friendly as authors claim.  This reviewer has just one suggestion.  Even though aouthors state that their compressed air-driven micropump does not requires the mechanical structure taking up a lot of space, a very small (palm-top size) micropump incorporated onto a microfluidic system have also been reported (Sasaki et al. Electrophoresis 33 (12), 1729-1735 (2012)).  Authors should cite the literature and claim the advantage of their method.  After addressing, this manuscript can be published.

Round 2

Reviewer 2 Report

Overall I don’t believe the authors really addressed my questions or took them very seriously. They clearly didn’t take the advice to do a complete re-write. The manuscript is still hard to follow. But with the edits they made, and my more careful reading to try to discern meaning where it’s unclear, I’m afraid this paper doesn’t meet basic standards I would expect for conduct of experiments with proper controls and data analysis, nor is it clear what the advantages or novelty of this approach is. I would recommend rejection at this point, and I can not recommend another journal for which this paper would be acceptable without major changes to the experimental design.

First to some of the minor points that they addressed from my comments:

I appreciate that they got rid of the misleading term “self-driving micropump” but “compressed air-driven micropump” isn’t much better. An air-driven pump is a mechanical device where compressed air flows continuously through a device like a centrifugal pump or a reciprocating piston mechanism such that it is continuously transferring energy from the air to the fluid, and can be operated continuously at steady state. The mechanism they describe is not really a pump at all. There is an initial input of work (compressing the air with the plunger) to store potential energy, and then the dissipation of that stored energy through expansion of the air pocket, which in turn does work on the fluid. This is by nature not steady-state; if you watch for a short enough period of time the flow rate will not change that much, but the underlying assumption is that you are only using a small portion of the stored energy. Rather than call it a “pump” they should call it what it is: pressurizing the fluid with a syringe. It’s a simple mechanism and it doesn’t require a fancy or misleading name.

The author’s claim that endpoint detection would commonly be done with a PMT, and that the “conditions of use of PMT are more stringent” than CMOS detection. I don’t know what that means – maybe they mean that PMTs are fragile or require high voltage. But for these droplets I think they’d do fine with a photodiode. It’s not like they need to do photon counting or ultra-low-light detection (the domain of PMTs), just distinguish positive (bright) from negative (less bright). Anyway, the question wasn’t really about what sensor to use. If they like using a CMOS sensor, they can still do that in endpoint mode by just counting the droplets one by one as they go past a particular spot. What benefit is there to being able to image the droplets across multiple turns of the fluid path?

They claim that “observing the real-time dynamics of the droplets” can “reduce the failure rate of the experiment”. What does that mean? There is no data in the paper to support that. How does “real-time dynamics of each droplet using CMOS observation” allow them to predict “ahead of the end point detection”? If by this they mean they are able to discern fluorescence changes at an earlier cycle than when they set their endpoint, they could just set a different endpoint. How does their system allow them to track droplets individually? And how do the experiments “fail” if they don’t do this? There is something in line 171-174 about “the fluorescent droplet counting software installed on PC” and drawing a “brightness map of the droplet through the derived Excel table to get the linear relationship”. This is all pretty vague and doesn’t explain how they manage to track a particular droplet through multiple turns of the tube around the mandrel, nor how that cycle-by-cycle data is used in the droplet format.

They now say they installed a pressure gauge on their syringe. This is not shown in Figure 1. The pressure gauge seems like an unnecessary complication, because the pressure will nicely follow the ideal gas law: compress to 1/2 the volume, double the pressure. The question is more about start-up. If they first compress the air pocket in the oil syringe, say to a pressure of 1 bar (e.g. by compressing a 10 mL air gap down to 5 mL), and then quickly thereafter compress the aqueous syringe also to 1 bar, during the few seconds that it takes to move from one to the other, oil will backflow into the aqueous syringe. The solution to this is easy, to put both syringes in a mount where you can use a bar to push both plungers simultaneously. They could do this with a common double-headed syringe pump without actually turning it on, for example. The other solution would be a check valve to prevent back flow. However, they don't describe any mechanism to synchronize applying pressure to both syringes at the same time, so I assume there is none.

I still fail to see how threading two 34 gauge needles (blunt, I hope, otherwise this is a huge safety hazard) into a PTFE capillary is in any way easy. I have done such operations myself in my younger days with a single blunt needle; now I wouldn’t attempt it due to my aging eyesight. Hot glue doesn't adhere very well to PTFE (nor do most other adhesives) so I find this a dubious method of sealing. If the cost of a commercial T or Y capillary fitting is too much, I question how they’re going to pay for PCR reagents. I can get a leak-tight “Y” fitting for capillary for $24 USD, no hot glue required, and you can easily adapt to other tubing sizes. I can get 100 reactions of PCR mix for about $100. It would cost me thousands of dollars to make the setup they describe. It would also cost me a lot to head to medical clinic if I poked myself with a needle, especially with a clinical sample containing hepatitis B virus.

Kidding aside, it also seems like a pretty low-throughput technique, if every single sample requires going through this needle-threading operation and gluing the needles to the capillary tube, and then 30 minutes or so to run each sample. I guess that means also that the capillary tube is single-use. So then for each experiment do you need to carefully wind a fresh Teflon capillary tube around the mandrel? Meanwhile I can run 96 samples by BioRad ddPCR in about 4 hours.

And now to where the paper ultimately fails to meet requirements for publication:

Figure 3 still does not show a no-template control, and as presented the PCR experiment is unsuitable for publication. A no-template control is absolutely critical both for their standard qPCR measurement in Figure 3a, and for setting their threshold as in Figure 3b. Having done ddPCR with the Bio-Rad instrument: you tend to get a cluster of negatives, and a cluster of positives, and a small number of droplets that have intermediate fluorescence. It is noticeable in Fig 3b that their positive droplets have a pretty wide range of values, but there isn’t a clear cluster of positives. In the terminology of Bio-Rad, all of the droplets that are above what look like the main cluster of negatives are what would be termed “rain”, i.e. droplets that fall at intermediate values between the “high” and “low”. I would recommend re-drawing Figure 3b as a histogram (brightness on X axis, frequency or counts on Y axis) for a reaction with template (perhaps the droplet population currently shown in Figure 3b) and a reaction with no template. I would then recommend repeating this sort of experiment many times (at least each 10 for positive control and no-template control) to see how reproducible the distributions are.

Line 266-272: so the copy number of DNA is being calculated from the droplet experiment? However it is not 18.16, 127.92 and 857.00. These are too many significant figures. IF we believe the droplet measurement is precise, it might be possible to say something like 18 ± 3, 130 ± 10, 860 ± 50. As the authors point out in their response to me, they are correcting their results by assuming droplet occupancy follows a Poisson distribution. This would suggest they have sufficient knowledge to realize that sampling statistics are also at work in the amount of DNA from the original sample that gets input into their droplet experiment, and into their qPCR experiment. As described above, if they repeat their experiment 10 times, they will calculate 10 different copy numbers, owing to sampling statistics (if everything else were perfectly reproducible). Each time they draw 10 uL of sample out of their stock of template, they actual number of template copies will follow a Poisson distribution.

The R^2 in text is 0.9981 whereas figure 3C says it’s 0.9995. However based on 3 data points they might as well say R^2 = 1.0. Normally when one does qPCR to develop a standard curve, one does at least 3, preferably more, replicates at each concentration, and preferably a wider range of concentrations, perhaps 6-8 concentrations spanning a range of 10^4-fold or more. One can then generate a standard curve that has confidence intervals. I see no attempt to do that here. As presented this PCR experiment is unacceptable for publication.

The larger issue here however is that it appears they are using a novel method (the capillary droplet dPCR method) to establish a quantitation that they then feed into what should be the gold-standard (qPCR). This experiment is  meaningless. If we take their standards as 10^5, 10^4, and 10^3 IU/mL as truly being 10-fold differences in concentration, they ought to get a log-linear relationship between sample concentration (be it in IU or copy number or whatever) and Ct. What is truly lacking here is an independent quantification of the DNA standard. I would recommend getting something that has been quantified and validated (e.g. one can purchase various quantitative DNA and RNA standards that have been validated by an established ddPCR method); use this to establish the validity of the novel dPCR method. The role of the qPCR is unclear hear. They ought to be able to take a DNA standard and perform a qPCR standard curve; this would for example allow them to cross-validate a measurement they with their novel dPCR on an “unknown” sample, but as presented in this paper, the qPCR is just a separate measurement they make that has no bearing on validating the dPCR measurement.

If, despite my recommendation for rejection, this manuscript were to recalled for revision, it still requires extensive editing for English, and I would still recommend a complete re-write. Just pulling some examples from the first page:

Line 13: “monitory” should be “monitoring”

Line 14: “pipeline” is not the right word here – generally “pipe” connotes something big. Small diameter like this is “tubing”, “capillary” or “microchannels”.

Line 21: “clinical-level” serum sample – what does “clinical-level” mean? Side  note, I didn't see anything in the methods or results about clinical samples or serum, just some standard of 10^5, 10^4, 10^3 IU/ML (although methods/results don't state, I guess it could be HBV DNA because that's often measured in IU).

Line 23-24: “Because the system greatly simplified…” This sentence doesn’t make sense.

Line 29: PCR has not just been “proposed” for 30 years, it’s been widely used and practiced, and it’s now more like 37 years.

And some others that caught my eye:

Line 83: droplets don’t “melt” because they’re already liquid. I think the word they are going for is either “coalesce”, or “break apart” (depending on which phenomenon they are worried about; “melt” is unclear).

Line 84: “the product of each cycle is clearly understood” – what does this mean?

Line 102: “Micro-pipeline” – just call it a microtube or capillary

Line 107: earlier they called the tube material Teflon (brand name) and here they call it by the polymer name PTFE. Choose one and stick with it.

Line 179-180: “no matter how the electric pump is modified it cannot solve the defect that the mechanical structure takes up a lot of space” – please avoid pejorative language. Have they heard of electrokinetic pumps? Side question, but can they explain in their application why space or power supply is at a premium? Are they proposing to do this in a remote setting like point-of-care? How do you then power the camera, or the hot glue gun to seal the capillary closed?

Line 243: If the droplets have a length of 0.5 mm, they don’t have a volume of 8.83 nL – you can’t get 3 significant figures in volume from a length measured to one significant figure. The volume is 9 nL.

Author Response

This manuscript is a resubmission of an earlier submission. The following is a list of the peer review reports and author responses from that submission.

Round 1

Reviewer 1 Report

The authors describe a passive flow driven droplet generator to perform real time digital PCR analysis. While such a platform is quite useful for making techniques like digital PCR portable and more accessible, I think the current manuscript lacks a clear description of "new" research component. I list my concerns below.

This work is based on the authors previous publication (Anal. Chem. 2018, 90, 20, 11925–11932). That 2018 paper had demonstrated the passive fluid pumping mechanism and applied for real time detection. Furthermore, the procedure and mechanism to generate droplets without the need of any syringe pumps quite novel and useful for other researchers. In the current manuscript, the authors fail to describe how this work is different from their 2018 publication. They mention "novel" many times including the abstract but I was unable to see the novelty here. For example, in the abstract it says "for the first time in the world, our system has realized the real-time detection of droplet fluorescence in each thermal cycle . . . " . This has already been demonstrated in the 2018 paper.

The authors should have their manuscript revised by a native English speaker/writer to remove all grammatical and technical mistakes. For example: line 50 should read "water and oil phase". . . Line 47 should read "technology, where" . . . Line 116 ul should read μL  . . . etc.

In the current manuscript, it is not clear how the "wire" in figure is maintaining the pressure gradient required to drive the fluid. They should briefly describe the mechanism in this paper. 

Reviewer 2 Report

The reviewer feels that the dPCR system proposed in this work is about the same as the system proposed in “Miniaturized Continuous-Flow Digital PCR for Clinical-Level Serum Sample Based on the 3D Microfluidics and CMOS Imaging Device”, which was conducted by the same group of authors and not cited in this paper. They use the same microfluidic device connected with 34g needles, nearly the same thermal cycling procedure, and the same real-time fluorescence monitor. 

The only difference found by the reviewer is that the syringe pump used in this work is “self-driven”, realized by compressing air into the syringe pump. From the reviewer’s view, the few new contributions are not sufficient for publication.